# Analysis of Citrus Tristeza Virus Incidences within Asian Citrus Psyllid (*Diaphorina citri*) Populations in Florida via High-Throughput Sequencing

**DOI:** 10.3390/insects13030275

**Published:** 2022-03-10

**Authors:** Kellee Britt, Samantha Gebben, Amit Levy, Diann Achor, Peggy Sieburth, Kristian Stevens, Maher Al Rwahnih, Ozgur Batuman

**Affiliations:** 1Department of Plant Pathology, Southwest Florida Research and Education Center, University of Florida, Immokalee, FL 34142, USA; kbritt2005@ufl.edu (K.B.); samantha.gebben@ufl.edu (S.G.); 2Department of Plant Pathology, Citrus Research and Education Center, University of Florida, Lake Alfred, FL 33850, USA; amitlevy@ufl.edu (A.L.); dsar@ufl.edu (D.A.); peggy.sieburth@ufl.edu (P.S.); 3Department of Evolution and Ecology, University of California-Davis, Davis, CA 95616, USA; kastevens@ucdavis.edu; 4Department of Plant Pathology, University of California-Davis, Davis, CA 95616, USA; malrwahnih@ucdavis.edu

**Keywords:** *C*Las, *Diaphorina citri*, insect-specific viruses, HTS analysis, Tristeza disease

## Abstract

**Simple Summary:**

The iconic citrus crop of Florida is suffering from an incurable disease called Huanglongbing (HLB), or citrus greening. HLB is caused by a bacterium *Candidatus* Liberibacter asiaticus (*C*Las), which is efficiently vectored by a small insect, the Asian citrus psyllid (ACP). To control the spread of *C*Las and ultimately HLB, citrus growers primarily depend on environmentally harmful insecticides against the ACP, which can induce insecticide resistance. An alternative, biological approach involves insect-specific viruses (ISVs) of the ACP for future exploitation to exclusively target this insect pest. To identify these ISVs, we have subjected ACPs to high-throughput sequencing, a powerful sequencing technology. This current study follows an initial detection of citrus tristeza virus (CTV) amongst the viruses detected in the ACP during HTS and continues to show the persistent and diverse presence of CTV in Florida ACPs across subsequent years. We suggest that the ACP may be utilized as a tool or lens of CTV presence throughout Florida citrus groves.

**Abstract:**

The destructive citrus disease, Huanglongbing (HLB) or citrus greening, continues to devastate Florida’s citrus industry. A hemipteran insect, the Asian citrus psyllid (ACP), disperses *Candidatus* Liberibacter asiaticus, one of the putative bacterial pathogens of HLB. This study builds upon ongoing research utilizing high-throughput sequencing to analyze the virome of ACP populations collected from citrus groves throughout Florida. Following the widespread detection of sequences aligning to the genome of citrus tristeza virus (CTV) across consecutive years in the Florida ACP virome, we continued to detect a pervasive amount of CTV in Florida ACPs during subsequent years. Simultaneously, we also detected mixed infections of CTV strains in pooled ACPs from different Florida regions. Predating the HLB epidemic, CTV has been present in Florida for many years and our results confirm its widespread and diverse persistence in Florida citrus groves through a unique lens, the ACP. CTV presence in the ACP likely results from feeding on CTV-infected citrus trees in Florida citrus groves, which may help to understand an overlapping presence of CTV and HLB, both endemic citrus pathosystems in the state, and their role in future integrated pest management strategies.

## 1. Introduction

Florida citrus groves face overwhelming devastation from the disease Huanglongbing (HLB) or citrus greening. This deadly citrus disease in North America, presumably caused by the bacterium *Candidatus* Liberibacter asiaticus (*C*Las), is vectored by the Asian citrus psyllid (ACP; *Diaphorina citri*) [1,2]. Citrus currently lacks resistance to this disease and management relies primarily on controlling ACP populations in groves and the use of tolerant citrus varieties [3]. The ACP and HLB are now endemic to Florida, with more than 95% of trees in the state affected. Reducing dependence on environmentally harmful insecticides to control the ACP motivates research for identifying potential biological alternatives. Such biological controls include microorganisms naturally present in the ACP, specifically viruses known as insect-specific viruses (ISVs) [4,5]. Researchers in Florida began identifying a potential ISV associated with the citrus pest early in the establishment of HLB in the state [6]. Nouri et al. [4] and subsequent studies expanded ISV analyses in the ACP using metagenomics of worldwide ACP populations and identified several novel ACP-associated viruses [7,8]. High-throughput sequencing (HTS) has the powerful and highly sensitive capacity to detect novel viruses in insects, along with other components of their microbiome [9]. However, a limited population of Florida ACPs was included in these studies, which inspired our previous investigation [10]. The endemic presence of HLB and the ACP in Florida citrus groves prompted a need to apply HTS solely to field populations of Florida ACPs originating from commercial citrus groves throughout the state with a goal to identify potential ISVs suitable for biological control of the ACP [10].

By applying HTS to representative populations of Florida ACPs, we detected ISVs associated with ACPs of this region as well as the widespread presence of viral sequences belonging to the citrus tristeza virus (CTV) genome across consecutive years [10]. CTV is a citrus phloem-limited closterovirus, thus allowing for uptake during phloem-feeding by the ACP [11]. Since its first detection in Florida almost 70 years ago, CTV has constantly posed a threat to the state’s citrus industry and continues to be the major viral pathogen to the crop throughout the world [11,12,13]. The shift away from planting citrus groves on the CTV-susceptible sour orange (*Citrus* × *aurantium*) rootstock has greatly decreased the susceptibility of trees to CTV; yet, as HLB continues to devastate Florida citrus groves, some growers are considering switching back to more *C*Las-tolerant rootstocks, such as sour orange and its hybrids [14,15,16,17]. The detection of a wide number of sequences aligning to the CTV genome suggests the persistence of the virus in Florida citrus trees, perhaps without causing discernable symptoms in tolerant trees. This may be due to the dominant presence of HLB symptoms in trees across Florida. Past surveys for CTV in Florida have shown the diverse and mixed presence of CTV strains throughout the state, predominantly the strains T36, VT, and T30 [18,19,20].

To further investigate the continued widespread and persistent presence of CTV in the virome of Florida ACPs, this current study applied Britt et al.’s [10] HTS approach but separated samples by year. Here, we report our survey results and discuss the importance and their potential implication in the development of a regional integrated pest management (IPM) strategy in citrus. We also discuss how this can be a unique view of endemic citrus pathosystems throughout Florida.

## 2. Materials and Methods

### 2.1. Insect Collection for RNA Extraction and HTS Analysis

ACP collections and RNA sample preparation for this study were based on the materials and methods of Britt et al. [10]. There were two groups of samples in this study, “Group 2019” and “Group 2020” (i.e., ACPs collected during 2019 and 2020, respectively). Each group had five samples designated for sequencing (i.e., HTS libraries). For Group 2019, samples were from five larger regional composite samples of 15 major citrus-producing counties in Florida as listed in Table 1. Specifically, counties for Region A included Lake, Orange, and Seminole; Region B, Pasco, and Polk; Region C, De Soto, Hardee, and Manatee; Region D, Martin, Okeechobee, and St. Lucie; Region E, Charlotte, Hendry, Lee, and Collier. Subsequently, total RNAs were extracted from pools of ACPs (~5–10 nymphs and/or adults from each county per region, *n* = ~100) collected from May to June of 2019. County ACPs were then combined into their larger regional sample (Table 1).

For Group 2020, 18 major citrus-producing counties were included as: Region A, Lake, Orange, Osceola, and Seminole; Region B, Pasco, and Polk; Region C, De Soto, Highlands, Manatee, and Sarasota; Region D, Indian River, Okeechobee, and St. Lucie; Region E, Charlotte, Glades, Hendry, Lee and Collier (Table 1). Total RNAs of Group 2020 were extracted from ACPs collected from April to June of 2020 (*n* = ~180).

### 2.2. High-Throughput Sequencing (HTS) Assembly and Analysis

Total RNAs were extracted from ACPs of Group 2019 and Group 2020 using TRIzol™ Reagent (Thermo Fisher Scientific, Waltham, MA, USA) according to the manufacturer’s instructions. Each year had five samples, one for each regional composite of counties previously mentioned (*n* = 10 libraries). Following extraction, each sample was quantified and evaluated for sufficient purity and concentration for HTS using a Synergy HTX plate reader (BioTek Instruments, Winooski, VT, USA). Both groups of total RNAs were delivered to Foundation Plant Services located at the University of California, Davis. Aliquots of the composite samples for both groups were subjected to the same HTS preparation and sequencing on the Illumina NextSeq 500 platform as previously described [21]. Resulting Illumina reads of 75-base length from all groups of cDNA libraries were demultiplexed, adapter trimmed, and filtered using Illumina’s bcl2fastq software version 2. Subsequently, trimmed single-end reads were de novo assembled into contiguous consensus sequences (contigs) using SPAdes version 3.14 with default parameters [22]. To annotate known viruses, de novo assembled contigs ≥200 bases were compared against a July 2020 copy of the GenBank database from the National Center for Biotechnology Information (NCBI) using BLASTn v. 2.10.1 with a word size of 7 and default parameters [23]. Finally, for more sensitive identification of viruses within the scope of this study, trimmed reads were aligned to the viral division of the GenBank nucleotide database using bowtie version 2.4.2 [24] and subsequently processed through PathoScope version 2.0.6 [25] for absolute read counts for each virus. The NCBI GenBank Accession numbers used for PathoScope, were: Diaphorina citri associated C virus (DcACV), KX235518 and KX235519; Diaphorina citri flavi-like virus (DcFLV), KX267823; Diaphorina citri densovirus (DcDV), KX165268; *Wolbachia* phage (WO), KX522565; and Diaphorina citri reovirus (DcRV), MT027144-MT027153 [4,7,8,26]. Virus genome fold coverage of contigs was calculated as the total contig length divided by the virus genome length (RefSeq genome length used if multiple genomes exist). Viral read counts were normalized as reads per kilobase million (RPKM) sequenced. RPKM was then used to compare the relative abundance of each virus. For CTV, the NCBI Reference Sequence (RefSeq) genome length of 19,296 nucleotides (nt) was used (GenBank Accession Number: NC_001661).

To further characterize the read coverage and distribution profile of the CTV genome, the software Geneious Prime 2021 v. 0.3 was utilized [27]. CTV genomes identified through PathoScope analysis were chosen based on their maximum mapped read coverage for each sample. A read was counted in the PathoScope analysis if it aligned to one of multiple CTV references. To generate such a coverage profile, a single best reference was required. The reference with the highest mapped coverage was selected and all trimmed reads were remapped to this reference. Some reads (an average of ~21% per sample) were missing from this subsequent analysis due to divergence from the selected reference genome.

### 2.3. Statistical Analysis

To statistically analyze the differences between the relative abundance (RPKM values) of the viruses of interest detected in each group, we used JMP^®^ 15 (SAS Institute Inc., Cary, NC, USA). The RPKM values of all regions were combined for each virus in each group for mean virus RPKM values. The virus RPKM means did not satisfy assumptions (normality and homoscedasticity) for analysis of variance (ANOVA) and were log-transformed with a constant of 1 added to all RPKM values before the transformation. They were then subjected to the nonparametric Kruskal-Wallis test [28]. If significance was found at *p* < 0.05, then data were subjected to the post hoc Dunn’s test [29] to determine which virus RPKM means significantly differed between each other within each group.

### 2.4. Validation of CTV Presence in HTS Samples and Field ACPs by RT-PCR

To further validate the presence of CTV in the HTS samples and randomly selected RNAs extracted from Florida field-collected ACPs not included in HTS, we utilized reverse-transcription PCRs (RT-PCRs). For cDNA synthesis, six microliters of total RNA and 1.5 µL of a random hexamer primer (250 ng/µL) (Thermo Fisher Scientific, Waltham, MA, USA) were incubated at 65 °C for 5 min followed by incubation on ice for another 5 min as previously described [10]. One µL of Superscript II (200 u) reverse transcriptase (Invitrogen, Carlsbad, CA, USA), 1 µL of dNTPs (10 µM), 2 µL of DTT (0.1 µM) (Thermo Fisher Scientific, Waltham, MA, USA), 4 µL of buffer (5×) (Thermo Fisher Scientific, Waltham, MA, USA) and 4.5 µL of RNase-free H₂O were then added in a total reaction mix of 20 µL and incubated at 42 °C for 1 h. The cDNA was then heated to 65 °C for ~15 min to deactivate Superscript II. The ACP cDNA was immediately subjected to RT-PCRs targeting the CTV coat protein (CP) [30]. The primers were Forward: 5’-ATG GAC GAC GAA ACA AAG AAA TTG-3’, and Reverse: 5′-TCA ACG TGT GTT GAA TTT CCC A-3′, which generated a 671-bp amplicon of the CTV CP gene [30]. PCRs were performed in a C1000 Touch Thermal Cycler (Bio-Rad Laboratories, Hercules, CA, USA) with the following PCR conditions: initial denaturation at 95 °C for 3 min, and 35 cycles of 95 °C for 30 s, 59 °C for 30 sec, and 72 °C for 1 min, with a final extension of 72 °C for 5 min. Subsequent PCR products were separated on 1% agarose gels, visualized by staining with Apex™ Safe DNA Stain (Genesee Scientific, San Diego, CA, USA), and selected PCR amplicons were Sanger sequenced and verified.

### 2.5. CTV Strain Detection in Florida ACPs by Strain-Specific RT-PCR

Total nucleic acid extraction from ACP samples was accomplished by the SDS/potassium acetate method as previously described [31]. CTV strains were detected in ACP using TaqMan RNA-to-Ct 1-Step-PCR kit (Applied Biosystems, Foster City, CA, USA) according to the manufacturer’s instructions. Strain-specific RT-PCR was conducted according to the TaqMan RNA-to-Ct 1-Step-PCR kit instructions, using strain-specific primers and probes as previously described by [32].

### 2.6. Immunogold Labelling of CTV Virions

Virion purification from Florida field-collected ACPs was performed as previously described with modifications [33]. Specifically, 250 μL of 0.01 M phosphate buffer (pH 7.0) (Thermo Fischer Scientific, Waltham, MA, USA) was added to ~0.5 g of ground ACP tissue (ACPs collected from Southwest Florida Research and Education Center citrus groves) and diluted in 1:1 (*v*/*v*) with (1X) TBE (Tris-borate-EDTA) buffer (Thermo Fischer Scientific, Waltham, MA, USA). The ACP/buffer mixture was then vortexed until all tissue was suspended, followed by centrifugation at 11,000 rpm (Eppendorf™ 5424 Microcentrifuge, Hamburg, Germany) for 2 min. After centrifugation, the supernatant was then transferred into a new 1.5 mL Eppendorf tube. Vortexing and centrifugation were repeated three times. All three volumes of supernatant were collected and filtered through a 33 mm filter unit of 0.45 μM (Millex^®^ Millipore Sigma, Sigma Aldrich, St. Louis, MO, USA) and then placed onto a nine-milliliter-thick 30% sucrose layer in a polycarbonate tube (Beckman Coulter, Brea, CA, USA). Each tube was then centrifuged in a Sorvall Discovery 90SE High-Speed Centrifuge (Kendro, Newtown, CT, USA) at 38,000 rpm using a Rotor type SW 50.1 (Beckman Coulter, Brea, CA, USA) for four hours. The resulting pellet was resuspended in 50 μL of 0.01 M phosphate buffer (pH 7.0) (Thermo Fischer Scientific, Waltham, MA, USA). This purified prep was placed on a 400-mesh carbon/formvar coated grid and rinsed with bacitracin water. After blocking for 15 min. in 1% BSA/0.1% triton X phosphate buffered saline (PBST) pH 7.2, the prep was incubated for 1 h with an antibody against the major CTV coat-protein (CPm) 1:100 in 1X PBS. The prep was rinsed three times with 0.1% PBST and incubated for 30 min. in GAR 20 nm gold (1:50 in 1X PBS). Prep was rinsed with PBS and bacitracin water and then negatively stained with 1% Ammonium Molybdate. Viral particles were then photographed on the Morgagni 268 transmission electron microscope (FEI, Hillsboro, OR, USA).

## 3. Results

### 3.1. HTS Generated a Persistent Presence of CTV Contigs in Total RNAs Extracted from Florida ACPs

High-throughput sequencing (HTS) of the ten Florida regional composite samples of total RNAs generated an average of approximately 33 million reads per cDNA library and an average of 52,837 de novo assembled contigs ≥200 nts long (Table 2).

As the regional HTS samples of Group 2019 lacked some counties included in Group 2020 and vice versa based on availability, we looked at each group separately. We analyzed the average percent identity (% ID) of the contigs, their total length (all contig lengths added together) in the regional sample, and their genome fold coverage (X) of the respective virus for a more in-depth interpretation (Table 3). In Group 2019, the average % ID of contigs annotated as CTV was always >98% and had a wide range of total contig length between 4680 nt to 27,369 nt (Table 3). The total contig length of 27,369 nt represents multiple overlapping contigs hitting CTV genomes in BLASTn and the total of their lengths. Genome coverage by the CTV contigs also had a wide range from 0.24X to 1.42X (Table 3). We analyzed the relative abundance (RPKM) of CTV and other viruses of interest for both groups (nontransformed data listed) to compare the diverse presence of the viruses detected in the Florida ACPs. For Group 2019, Region C had the lowest relative abundance of CTV, while Region B had the highest relative abundance of CTV (Table 4). Along with read count, PathoScope also identified CTV strains with the most mapped CTV read coverage, and identified three different strains, T36, resistance-breaking (RB), and VT in Group 2019 (Table 4). Based on the Kruskal–Wallis test and Dunn’s test of the RPKM means, significant differences between the relative abundance of the viruses were observed in Group 2019, specifically the ACP-associated DcACV relative abundance was significantly higher than transcripts detected from the bacteriophage WO (Table 4). It is important to note that we looked at the transcriptome of the samples and DNA virus sequences are regarded as expression.

For Group 2020, the average % ID of contigs annotated as CTV was always >97%, with total contig lengths ranging from 276 nt to 9518 nt, and 0.01X to 0.49X genome coverage (Table 5). Region D had the highest relative abundance of CTV, while Region C had the lowest relative abundance of CTV (Table 6). Regarding CTV strains identified with most mapped read coverage, T36 was the most common, like Group 2019, but T30 was also detected, along with VT (Table 6). Significant differences between the relative abundance of each virus in Group 2020 were observed, specifically that DcACV was significantly higher than DcFLV, DcRV, and WO (Table 6).

In addition to CTV, HTS detected known ISVs associated with the ACP, specifically DcACV, DcFLV, DcDV, and DcRV (Table 3, Table 4, Table 5 and Table 6). DcACV showed high abundance compared to the other viruses for both groups (Table 3, Table 4, Table 5 and Table 6). HTS also detected transcripts of WO in Regions B and C of Group 2019 and Region D of Group 2020 (Table 3, Table 4, Table 5 and Table 6). WO annotated contigs had high % ID, but very low fold coverage of the genome (Table 3 and Table 5), which suggests low expression of the phage and was consistent with its low relative abundance for most of the HTS samples (Table 4 and Table 6).

### 3.2. CTV Read Coverage Profiles of HTS Samples

By mapping each HTS sample’s CTV reads to the genome selected by PathoScope as having the highest mapped read coverage, we sought to characterize genome wide CTV read distribution and to identify consistent regions of read coverage between the samples. For all regions, there were genome coverage gaps and it appeared that 9–18 kilobase (kb) CTV genome region showed the highest coverage of reads (Figure 1 and Figure 2). The HTS samples showed very little read coverage at the 5′-end of the CTV genome but increased in coverage towards the latter half of the CTV genome.

### 3.3. RT-PCRs of Florida ACPs

RNA extracts from the HTS samples and extracted RNAs from random samples of ACPs not included in HTS analysis were screened using RT-PCRs of the CTV CP to validate HTS findings of widespread CTV. RT-PCRs for the ten HTS samples were done in quadruplicate (*n* = 40) and tested positive for the CTV CP gene ~68% of the time (data not shown). Positive incidences in additional ACP RNAs (*n* = 53) that were not included in HTS were ~50%, which provides further evidence of the widespread presence of CTV in HTS samples and Florida ACP populations (data not shown). To further analyze the diversity of CTV strains in Florida ACPs, we also subjected RNA extracts from separate ACP populations to CTV strain-specific RT-PCRs. In the four samples analyzed, the T36 strain was always present, and the VT strain was present only in mixed infections with other CTV strains (Table 7).

### 3.4. Immunogold-Labelling Detects the Presence of Large CTV Particles in Florida ACPs

To corroborate the genetic findings of CTV and to further show the presence of CTV in Florida ACP populations, we extracted virions from ACPs collected from Florida citrus groves during 2017–2018 [10]. Using antibodies against the major coat protein (CTV-CP), long, flexuous rod-shaped, immunogold-labeled fragmented particles of CTV from the ACPs were detected (Figure 3).

## 4. Discussion

We initially intended to identify novel insect-specific viruses (ISVs) in Florida ACP populations by utilizing HTS analysis of the insects originating from commercial citrus groves throughout Florida across consecutive years [10]. An unanticipated discovery during this initial analysis was the widespread presence of CTV sequences detected throughout the Florida ACP virome. This finding regarding CTV prompted us to conduct a more sensitive analysis of the virome in Florida ACPs in subsequent years separately (i.e., 2019 and 2020). Similarly, during this study, HTS analysis of total RNAs extracted from ACPs collected during 2019 and 2020 showed the widespread presence of CTV, as well as the presence of several known ISVs associated with the ACP.

The low relative abundance of CTV compared to the ACP virus DcACV suggests CTV is likely a part of the citrus phloem contents consumed during ACP-feeding. We cautiously found that CTV T36 was the most common strain detected in the HTS samples in silico with mapped reads through PathoScope, nevertheless with coverage gaps. Interestingly, the CTV RB strain was detected in Region D of Group 2019, which is a strain that has not been detected in Florida before [34]. This, of course, would have to be further substantiated with strain-specific RT-PCR, as well as be similarly investigated in the citrus host itself, but alone presents a discovery worth mentioning. Though a majority of the HTS samples tested positive (~68%) for the CTV CP gene, the variable incidence of contigs throughout the samples (absence in Region C of 2019) suggest the low occurrence of CTV resulting from its presence as a part of the diet consumption of the ACP. This result and the finding of ~50% positive samples of ACP RNAs from field collected samples in Florida suggests varying CTV viral titers and fragments present in the total RNAs of the HTS samples and the Florida ACP virome in general. Nevertheless, the consistent presence of CTV across all the ACP samples regardless of the differences in counties between the regions and years, warrants consideration as a potential indirect indicator of widespread CTV in Florida.

The read coverage and distribution profile of the HTS samples against the CTV genome showed consistently gapped coverage, but higher coverage between 9–18 kb (Figure 1 and Figure 2). This region includes the RNA-dependent RNA polymerase (RdRp), proteins p33, p6, p61, the analog to heat shock protein (HSP70h), the minor coat protein (CPm) and major coat protein (CP) [11]. The significance of this region in the ACP is unknown but appears to differ compared to read coverage profiles generated from RNA-sequencing and viral sRNA read coverage profiles of CTV in citrus plants, which accumulated from the p13, p20, and p23 genes [35,36,37,38]. Transcriptomics of CTV-infected aphids have not been investigated and would offer an interesting comparison of CTV presence in the virus’s insect vector versus in a nonvector. Nonetheless, higher read coverage and distribution over the latter half of the genome could also be explained by the subgenomic and defective RNAs generated during the CTV replication process in the citrus host [39,40,41], and subsequent ACP acquisition of the RNAs during feeding. Both the gapped mapping results and low relative abundance demonstrate the presence of CTV likely as a part of the diet in Florida ACP.

The widespread presence of CTV strains in mixed infections (T36, T30, and VT), albeit in pooled ACPs from different Florida counties, is consistent with recent findings of the widespread and dynamic presence of CTV strains in mixed infections in citrus trees throughout the state, and the continued threat to Florida citrus groves [19,20]. Regarding diversity, we saw an average ~21% reduction in mapped reads compared to the absolute reads, which would suggest mixture of the strains that these absolute reads may have originated from (Figure 1 and Figure 2). From both PathoScope and strain-specific RT-PCR results, T36 was the most common strain detected, followed by VT (Table 4, Table 6 and Table 7). Immunogold-labelling of the major coat protein of CTV illustrated the presence of large CTV particle fragments in Florida ACPs, yet the presence of complete CTV virions remains to be determined.

Moreover, apart from CTV detection in Florida ACPs, the regional groups of HTS samples from each year in this study appear to complement the viral surveys conducted by Britt et al. [10] regarding the prevalence and dispersal of ACP-associated viruses, specifically DcACV, DcFLV, DcRV, and DcDV. This study continued to show the widespread and abundant presence of DcACV as initially shown by Britt et al. [10]. Although Britt et al. [10] and this study detected widespread CTV in Florida ACPs, these findings do not suggest that the ACP transmits CTV, but that the ACP likely consumes the virus when feeding on CTV-infected citrus trees. Our numerous attempts to transmit CTV from citrus to citrus with using ACPs, which was not part of this study, failed to show any evidence of a potential CTV transmission in controlled greenhouse conditions (Batuman et al., *unpublished data*). Furthermore, it is also not surprising that HTS detected contigs associated with the *Wolbachia* phage in Regions B, C, and D (Table 3, Table 4, Table 5 and Table 6) given the widespread distribution of *Wolbachia* and infection of the bacteria by the phage in arthropods [42,43].

## 5. Conclusions

Virome analysis of Florida ACP populations enabled a unique viewpoint of the decades-long presence of CTV in Florida citrus groves and showed an overlap with the HLB epidemic throughout the state. Results from this study continue to suggest that the perennial nature of citrus with CTV infection has allowed for endemic ACPs to uptake CTV particles during phloem-feeding on these trees in Florida groves. Nonetheless, our study clearly shows that CTV is still widespread in the main citrus producing regions throughout Florida, though potentially asymptomatic. Alternatively, symptoms in CTV-infected trees could be masked by more dominant symptoms caused by other widespread citrus diseases, such as HLB and/or citrus blight [44]. Furthermore, our study approach is a rather unique one, in the sense that we used an unconventional and indirect-survey approach in Florida citrus groves through use of a widespread phloem-feeding insect on citrus trees, the ACP. Strain-specific RT-PCR results show the likely presence of CTV in mixed infections in Florida ACPs, potentially with the *C*Las bacterium. These results offer a different angle regarding the presence and diversity of CTV throughout Florida citrus groves, but not a replacement of sampling directly from citrus hosts. It would be interesting to see how future research can compare the sampling of citrus trees for CTV presence and diversity versus pooled or individual ACPs within the same grove or greenhouse, etc. Nevertheless, results from this current study can also provide valuable information to the citrus industry (i.e., breeders, horticulturists, pathologists, growers, etc.) to consider the continued widespread presence of CTV (and its aphid vectors) and to avoid the use of CTV-susceptible rootstocks in the integrated pest management program(s) being developed for HLB control in Florida citrus groves.

## Figures and Tables

**Figure 1 insects-13-00275-f001:**
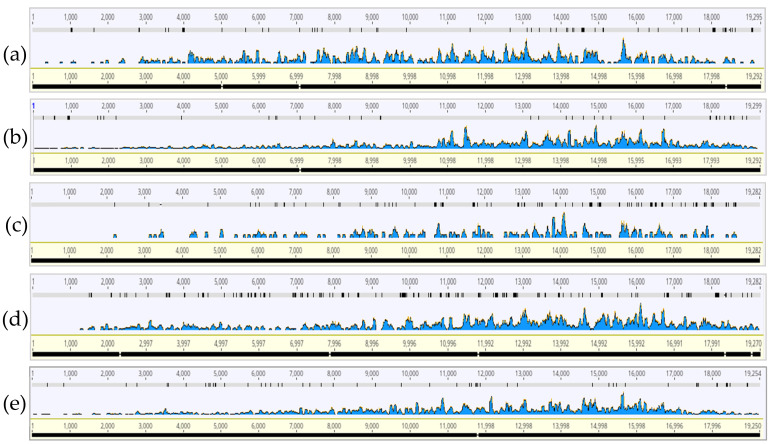
Read coverage and distribution profile generated from each Florida Region of 2019 aligned to the respective citrus tristeza virus (CTV). (**a**) Region A (CTV T36, KC517485.1); (**b**) Region B (CTV T36, KC517485.1); (**c**) Region C (CTV T36, KC517488.1); (**d**) Region D (CTV RB, KU358530.1); (**e**) Region E (CTV VT, KC748392.1).

**Figure 2 insects-13-00275-f002:**
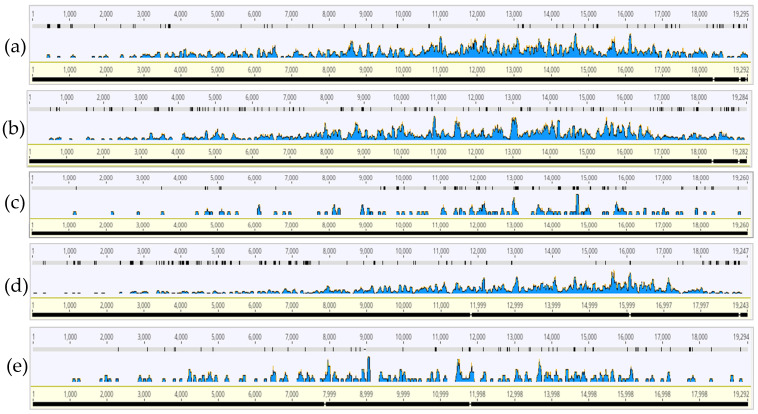
Read coverage and distribution profile generated from each Florida Region of 2020 aligned to the respective citrus tristeza virus (CTV). (**a**) Region A (CTV T36, KC517485.1); (**b**) Region B (CTV T36, KC517488.1); (**c**) Region C (CTV T30, EU937520.1); (**d**) Region D (CTV VT, KY110738.1); (**e**) Region E (CTV T36, KC517485.1).

**Figure 3 insects-13-00275-f003:**
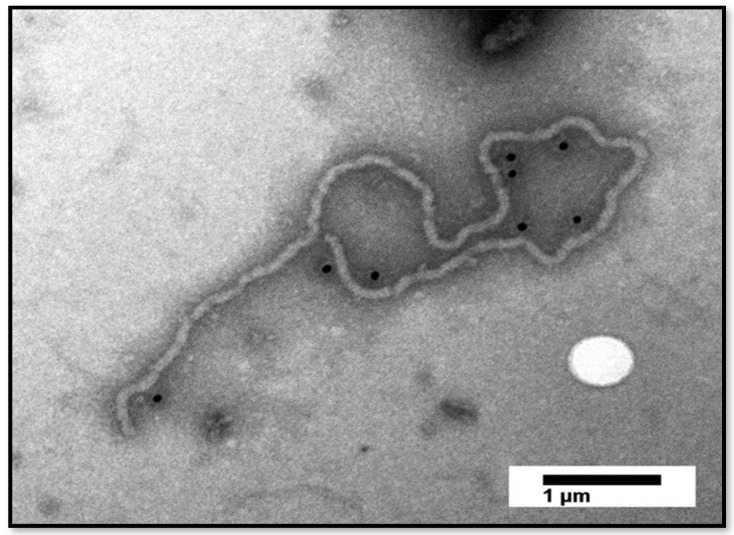
Transmission electron micrograph of immunogold-labeled citrus tristeza virus (CTV) particles purified from a composite sample of Asian citrus psyllids, using antibodies against the major coat protein (CTV-CP). Scale bar = 1 µm.

**Table 1 insects-13-00275-t001:** Composition of Florida regional Asian citrus psyllid (*Diaphorina citri*; ACP) samples per yearly group (five regions) compiled for total RNA extraction.

Group	Region	Florida Counties Included	The Approximate Number of Nymph and Adult ACPs Used for Composite
2019	A	Lake, Orange, Seminole	20
2019	B	Pasco, Polk	15
2019	C	DeSoto, Hardee, Manatee	20
2019	D	Martin, Okeechobee, St. Lucie	20
2019	E	Charlotte, Hendry, Lee, Collier	25
2020	A	Lake, Orange, Osceola, Seminole	30
2020	B	Pasco, Polk	20
2020	C	DeSoto, Highlands, Manatee, Sarasota	50
2020	D	Indian River, Okeechobee, St. Lucie	30
2020	E	Charlotte, Glades, Hendry, Lee, Collier	50

**Table 2 insects-13-00275-t002:** Summary of RNA-sequencing results generated from cDNA libraries of Groups 2019 and 2020 during HTS analysis.

Group	Region	Total Yield of Reads	Total Filtered Reads *	Assembled Contigs ≥200 nts
2019	A	2,687,116,425	35,828,219	52,675
2019	B	2,794,686,600	37,262,488	60,531
2019	C	1,545,277,950	20,603,706	45,251
2019	D	2,835,048,375	37,800,645	64,968
2019	E	2,997,707,400	39,969,432	51,178
2020	A	2,878,296,525	38,377,287	50,506
2020	B	2,909,774,550	38,796,994	59,991
2020	C	3,180,154,650	42,402,062	50,093
2020	D	1,611,839,475	21,491,193	48,713
2020	E	1,440,971,850	19,212,958	44,465

* The total number of reads after adapter-trimming and quality control (filtering) that were used for de novo assembling of contigs. The average read length was 75 nucleotides.

**Table 3 insects-13-00275-t003:** Viral contigs annotated as known viruses of interest for 2019 Florida Regions A–E.

Virus	Region A	Region B	Region C	Region D	Region E
CTV ^1^	99.38, 4680, 0.24X	98.25, 27,369, 1.42X	-	98.31, 7739, 0.40X	99.65, 12,742, 0.66X
DcACV ^2^	99.90, 7342, 1.75X	99.91, 7662, 1.82X	99.73, 4092, 0.98X	-	99.74, 4187, 0.99X
DcFLV ^3^	97.89, 24,970, 0.90X	98.89, 27,530, 0.99X	99.00, 1601, 0.60X	-	-
DcDV ^4^	-	84.63, 1438, 0.28X	84.63, 1382, 0.27X	91.24, 1659, 0.33X	-
WO ^5^	-	92.7, 305, 0.004X	92.39, 276, 0.004X	-	-

Average percent identity (% ID), total contig length (in nucleotides), and genome fold coverage are shown. ^1^ CTV: citrus tristeza virus; ^2^ DcACV: Diaphorina citri associated C virus; ^3^ DcFLV: Diaphorina citri flavi-like virus; ^4^ DcDV: Diaphorina citri densovirus; ^5^ WO: *Wolbachia* phage. The dash indicates no contig detection of that virus in that region likely due to a lack of adequate read coverage for contigs.

**Table 4 insects-13-00275-t004:** Abundance of viruses (with detected contigs of interest) in Group 2019 Florida Regions A–E.

Virus	Region A	Region B	Region C	Region D	Region E	Dunn’s Test *p* < 0.05 *
CTV ^1^	T36 ^6^ 1.00 (698)	T36 ^6^ 12.47 (8969)	T36 ^7^ 0.65 (259)	RB ^8^ 2.17 (1580)	VT ^9^ 3.26 (2518)	ab
DcACV ^2^	294.80 (44,287)	120.90 (18,887)	211.80 (18,298)	0 (0)	411.40 (68,947)	a
DcFLV ^3^	3.69 (3661)	47.79 (49,343)	0.81 (460)	0 (0)	0.09 (96)	ab
DcDV ^4^	1.16 (211)	0.85 (460)	0.91 (95)	0.96 (184)	0.81 (164)	ab
WO ^5^	0.09 (209)	0.09 (218)	0.12 (165)	0.04 (93)	0.06 (169)	b

The normalized reads per kilobase per million sequenced (RPKM) values, absolute number of viral reads in parentheses, CTV strain detected, and statistical analysis are shown. ^1^ CTV: citrus tristeza virus; ^2^ DcACV: Diaphorina citri associated C virus; ^3^ DcFLV: Diaphorina citri flavi-like virus; ^4^ DcDV: Diaphorina citri densovirus; ^5^ WO: *Wolbachia* phage; ^6^ KC517485.1; ^7^ KC517488.1; ^8^ KU358530.1; ^9^ KC748392.1; RB: resistance-breaking; * All data were log-transformed with an added constant 1 before statistical analysis. Kruskal-Wallis test, Chi-square: 9.6944, DF = 4, *p* = 0.0459. Viruses with different letters have significantly different normalized RPKM means of all regions based on Dunn’s test at *p* < 0.05.

**Table 5 insects-13-00275-t005:** Viral contigs annotated as known viruses of interest for 2020 Florida Regions A–E.

Virus	Region A	Region B	Region C	Region D	Region E
CTV ^1^	98.86, 7628, 0.40X	97.52, 9518, 0.49X	100, 276, 0.01X	97.99, 8445, 0.43X	100, 427, 0.02X
DcACV ^2^	98.60, 7551, 1.80X	99.62, 4166, 0.99X	95.35, 7238, 1.74X	99.80, 6021, 1.43X	99.95, 4133, 0.98X
DcFLV ^3^	-	-	98.36, 66, 0.02X	-	-
DcDV ^4^	84.63, 999, 0.20X	84.63, 1338, 0.60X	-	84.63, 989, 0.20X	84.63. 1033, 0.02X
DcRV ^5^	-	-	-	-	97.13, 19,515, 1.03X
WO ^6^	-	-	-	95.11, 554, 0.008X	-

Average percent identity (% ID), total contig length (in nucleotides), and genome fold coverage are shown. ^1^ CTV: citrus tristeza virus; ^2^ DcACV: Diaphorina citri associated C virus; ^3^ DcFLV: Diaphorina citri flavi-like virus; ^4^ DcDV: Diaphorina citri densovirus; ^5^ DcRV: Diaphorina citri reovirus; ^6^ WO: *Wolbachia* phage. The dash indicates no contig detection of that virus in that region likely due to a lack of adequate read coverage for contigs.

**Table 6 insects-13-00275-t006:** Abundance of viruses (with detected contigs of interest) in Group 2020 Florida Regions A–E.

Virus	Region A	Region B	Region C	Region D	Region E	Dunn’s Test *p* < 0.05 *
CTV ^1^	T36 ^7^ 1.62 (1200)	T36 ^8^ 2.17 (1622)	T30 ^9^ 0.21 (170)	VT ^10^ 4.38 (1817)	T36 ^7^ 0.71 (263)	ab
DcACV ^2^	165.51 (26,634)	208.69 (33,949)	51.33 (9126)	514.51 (46,364)	229.10 (18,458)	a
DcFLV ^3^	0.014 (15)	0.001 (1)	0.32 (372)	0 (0)	0.15 (78)	b
DcDV ^4^	1.16 (226)	0.84 (166)	0.44 (95)	0.68 (74)	0.78 (76)	ab
DcRV ^5^	0.001 (1)	0 (0)	0.02 (26)	0 (0)	514.19 (262,092)	b
WO ^6^	0.10 (241)	0.05 (131)	0.10 (268)	0.13 (190)	0.10 (131)	b

The normalized reads per kilobase per million sequenced (RPKM) values, absolute number of viral reads in parentheses, CTV strain detected, and statistical analysis are shown. ^1^ CTV: citrus tristeza virus; ^2^ DcACV: Diaphorina citri associated C virus; ^3^ DcFLV: Diaphorina citri flavi-like virus; ^4^ DcDV: Diaphorina citri densovirus; ^5^ DcRV: Diaphorina citri reovirus; ^6^ WO: *Wolbachia* phage; ^7^ KC517485.1; ^8^ KC517488.1; ^9^ EU937520.1; ^10^ KY110738.1; * All data were log-transformed with an added constant 1 before statistical analysis. Kruskal-Wallis test, Chi-square: 19.2101, DF = 5, *p* = 0.0018. Viruses with different letters have significantly different normalized RPKM means of all Regions based on Dunn’s test at *p* < 0.05.

**Table 7 insects-13-00275-t007:** Citrus tristeza virus (CTV) strains identified in Florida Asian citrus psyllid (*Diaphorina citri*; ACP) populations using strain-specific RT-PCR.

Florida County	ACP Sample (Number of Individuals)	Collection Date	Detected CTV Strains
Glades	Adult (5)	10/27/2017	T36, VT
Collier	Nymph (11)	11/1/2017	T36, VT
Orange	Adult (8)	3/9/2018	T36, VT
Collier	Adult (11)	3/13/2020	T30, T36, VT

## Data Availability

Additional data may be provided upon request.

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
