# Peer review of "Analysis of Citrus Tristeza Virus Incidences within Asian Citrus Psyllid (Diaphorina citri) Populations in Florida via High-Throughput Sequencing"

_insects, 2022, doi:10.3390/insects13030275_

Round 1
Reviewer 1 Report
This is an interesting paper, showing the nonselective uptake of CTV particles by nymphs and adults of Diaphorina citri (Dc) from the phloem tissue of infected plants. The paper raises some interesting questions that will be worth further study. e.g. Does the presence of CTV in HLB infected tissue and within the psyllid have an impact on the transmission C liberibacter asiaticus? Do the Dc benefit from the enrichment of their diet by CTV ?
The author's suggestion based on EM and reaction with antibodies to CTV coat protein is not sufficient to consider the particles as full length, simply because these antibodies do not recognize the c. 3 % virus end coated with the CP minor. This is a critical question since this CTV virion end is considered associated with aphid transmission . I suggest therefore to maintain the figure, only as an illustration of the presence of CTV particle segments of large size
Author Response
Dear Reviewer,
Thank you for your time and thoughtful reviews of our manuscript. We have considered and addressed (in bold) all of your valuable comments as below:
Reviewer 1
“The author's suggestion based on EM and reaction with antibodies to CTV coat protein is not sufficient to consider the particles as full length, simply because these antibodies do not recognize the c. 3 % virus end coated with the CP minor. This is a critical question since this CTV virion end is considered associated with aphid transmission. I suggest therefore to maintain the figure, only as an illustration of the presence of CTV particle segments of large size.”
We agree with this suggestion and have changed the parts in the manuscript describing the CTV particles detected in Florida ACPs as fragments and not the whole virion.
Reviewer 2 Report
This paper is a continuation of the previous publication (Britt et al. 2020), a good follow-up on the CTV issue with interesting discoveries. The approach was unique and straight forward in using HTS/metagenomic technology to investigate the viromes of ACP collected from Florida citrus groves. I have a few comments:
Line 137: Wolbachia phage (KO), KX522565. This is a DNA virus. In the method, RNA was extracted and analyzed. Is this meant that only the transcriptome of the phage was studied?
Table 2: Explain Total Filtered Reads.
Table 3 and 5: What are the “-“? No data? Why? Also, I felt the table should have better titles. The existing titles look like explanation notes.
Table 4: The table should be better organized. The column of Statistics should be in a note.
Line 374-375: CTV abundance is lower comparing to DcACV only.
Line 376: With so many coverage gaps (Figure 1 and 2), I am not sure it is the best to name the ACP CTV as strain T-36? I wonder the coverage in aphids using the same method.
Author Response
Dear Reviewer,
Thank you for your time and thoughtful reviews of our manuscript. We have considered and addressed (in bold) all of your valuable comments as below:
Reviewer 2
Line 137: Wolbachia phage (KO), KX522565. This is a DNA virus. In the method, RNA was extracted and analyzed. Is this meant that only the transcriptome of the phage was studied?
Yes, and this has been clarified (not implying genome detection) in the manuscript.
Table 2: Explain Total Filtered Reads.
The “Total Filtered Reads” were the number of reads following adapter-trimming that had also passed quality control filtering. This has been clarified in the manuscript.
Table 3 and 5: What are the “-“? No data? Why? Also, I felt the table should have better titles. The existing titles look like explanation notes.
There is a note underneath the table indicating that the dash line means that there were no contigs annotated as this virus in this sample. This was probably because of a lack of adequate reads and coverage to make these contigs (which has been clarified as a note). The titles of Tables are modified and improved.
Table 4: The table should be better organized. The column of Statistics should be in a note.
We agree with your suggestion and have made these changes.
Line 374-375: CTV abundance is lower comparing to DcACV only.
We agree with this statement and have corrected this in the manuscript accordingly.
Line 376: With so many coverage gaps (Figure 1 and 2), I am not sure it is the best to name the ACP CTV as strain T-36? I wonder the coverage in aphids using the same method.
We agree with this comment and have changed the wording in the manuscript to cautiously mention this statement since it there are substantial coverage gaps. We have not found any report describing the CTV coverage in aphid guts, but will be great to look at it in future experiments (thank you).
Reviewer 3 Report
Asian citrus psyllid (ACP) is of economical importance due to being the vector of Huanglongbing (HLB). Utilizing high-throughput sequencing to analyze co-existing viruses is of somewhat interest within ACP. However, a couple of years ago, Citrus tristeza virus (CTV) was reported to co-exist with HLB within ACP by other authors, the progress here is the proportion analysis of CTV populations within ACP populatins collected from Florida, which is a good work. It's more interesting to know the interaction between viruses and the pathogen of HLB, how can they become a unique lens of endemic citrus pathosystems, which might be needed to add extra experiment data, e.g. if CTV can be transmitted via ACP as well, or whether the titer change trend of Candidatus Liberibacter asiaticum should be monitored etc. Therefore, it is suggested not to strengthen "a unique lens of endemic citrus pathosystems".
BTW, there found a few format problems, which should be revised, e.g. H2O, scientific name of psyllid in the table.
Finally, it is suggested to change the title, either to omit "a unique lens of endemic citrus pathosystems" of the original title, or to directly describe the analysis of CTV populations within ACP populations in Florida via high-throughput sequencing, since virome analysis should also include analysis of other viruses.
Author Response
Dear Reviewer,
Thank you for your time and thoughtful reviews of our manuscript. We have considered and addressed (in bold) all of your valuable comments as below:
Therefore, it is suggested not to strengthen "a unique lens of endemic citrus pathosystems".
BTW, there found a few format problems, which should be revised, e.g. H2O, scientific name of psyllid in the table.
We agree with these suggestions and have changed these mistakes accordingly.
Finally, it is suggested to change the title, either to omit "a unique lens of endemic citrus pathosystems" of the original title, or to directly describe the analysis of CTV populations within ACP populations in Florida via high-throughput sequencing, since virome analysis should also include analysis of other viruses.
We agree with this suggestion and have changed the title accordingly.
Round 2
Reviewer 3 Report
The revised version looks good, which shows clearly the progress authors have made.